# The Effect of Supplemental Concentrate Feeding on the Morphological and Functional Development of the Pancreas in Early Weaned Yak Calves

**DOI:** 10.3390/ani12192563

**Published:** 2022-09-26

**Authors:** Yang Jiao, Shujie Liu, Yanan Zhou, Deyu Yang, Jilan Li, Zhanhong Cui

**Affiliations:** 1Qinghai Academy of Animal Husbandry and Veterinary Sciences, Qinghai University, Xining 810016, China; 2Yak Engineering Technology Research Center of Qinghai Province, Xining 810016, China; 3Key Laboratory of Grazing Livestock Animal Nutrition and Feed Science of Qinghai Province, Xining 810016, China

**Keywords:** supplemental concentrate feeding, yak calves, growth performance, pancreas development, metabolomics

## Abstract

**Simple Summary:**

This study aimed to investigate the nutritional intake deficiency on rearing yak calves. We investigated supplemental concentrate feeding effects on the morphological and functional development of the pancreas in early weaned yak calves. In the study, we determined the apparent digestibility of nutrients by digestion trail, the morphological development of the pancreas in yak calves by tissue sectioning, the activity of main digestive enzymes and hormone levels by ELISA kits, and the content of major small molecule metabolites in the pancreas by non-targeted metabolomics techniques. The morphological and functional development of the pancreas and its small molecule metabolites are mainly presented in graphical form, which had positive regulatory effects on the development of the pancreas in early weaned yak calves. In summary, we found that supplemental concentrate feeding was crucial for the high-quality growth and development of early weaned yak calves and had a positive influence on the intrinsic relationship between the overall development level and physiological functions of the pancreas, which could provide an important reference for scientific rearing of early weaned yak calves.

**Abstract:**

This experiment was conducted to investigate the effect of supplemental concentrate feeding on the pancreatic development of yak calves. Twenty one-month-old yak calves with healthy body condition and similar body weight were selected as experimental animals and randomly divided into two groups, five replicates in each group. The control group yak calves were fed milk replacer and alfalfa hay, the experimental group yak calves were fed milk replacer, alfalfa hay and concentrate. The pre-feeding period of this experiment was thirty days, the trial period was one hundred days. At the end of feeding trail, five yak calves from each group were selected and slaughtered and the pancreas tissues of yak calves were collected and determined. The results showed that: (1) Dry matter and body weight of yak calves in the test group were significantly higher than those of the control group. (2) The apparent nutrient digestibility of crude protein, crude fat, calcium and phosphorus in the test group of yak calves was significantly higher than that of the control group, while the apparent nutrient digestibility of neutral detergent fiber and acid detergent fiber in the test group was significantly lower than that of the control group. (3) Pancreatic weight, organ index, total ratio of exocrine part area and total ratio of endocrine area of yak calves in the test group were significantly higher than those in the control group, while the ratio of exocrine area was significantly lower in the test group than that of the control group. (4) The activities of the main pancreatic digestive enzymes: pancreatic amylase, pancreatic lipase, pancreatic protease and chymotrypsin were significantly higher in the test group than those of the control group, as were the hormonal contents of glucagon, insulin and pancreatic polypeptide. (5) The main differential metabolites of the pancreas in the test group were significantly higher than those of the control group, such as D-proline, hypoxanthine, acetylcysteine, gamma-glutamylcysteine, thiazolidine-4-carboxylic acid, piperidinic acid, ellagic acid, nicotinamide, tropolone, D-serine, ribulose-5-phosphate, (+/-)5(6)-epoxyeicosatrienoic acid(EET), 2-hydroxycinnamic acid, L-phenylalanine, creatinine, tetrahydrocorticosterone, pyridoxamine, xanthine, 5-oxoproline, asparagine, DL-tryptophan, in-dole-3-acrylic acid, thymine, trehalose, docosapentaenoic acid, docosahexaenoic acid, fatty acid esters of hydroxy fatty acids(FAHFA) (18:1/20:3), fatty acid esters of hydroxy fatty acids(FAHFA) (18:2/20:4), adrenic acid and xanthosine. In conclusion, supplemental concentrate feeding promoted the good development of morphological and functional properties of the pancreas in early weaned yak calves to improve the digestion and absorption of feed nutrients, so as to enhance the growth and development quality of early weaned yak calves.

## 1. Introduction

As the dominant livestock and poultry material raised by herdsmen in Qinghai Plateau, yak has extremely high biological and economic value [1]. As the basis of high-quality development of yak industry, the early cultivation of calves is very important for later growth and development [2]. The traditional way of breast-feeding may result in low immunity and slow growth and development of yak calves, resulting in insufficient nutritional intake of yak calves in breast-feeding and grazing conditions [3]. Due to the excessive influence factors under grazing conditions, Guan Jiuqiang et al. found that the body weight of yak calves was significantly higher than that of traditional grazing calves [4], which greatly promoted the growth and development of yak calves, and also improved the reproductive efficiency of female yaks. On the basis of early weaning, Chai et al. found that the weight of yak calves increased significantly from 60 days old to 180 days old, which improved the production performance of yak and the economic benefits to local herdsmen [5]. Sun found that supplementation of concentrate in the warm season significantly improved the average daily gain and slaughter performance of growing yaks under grazing conditions [6] and promoted the growth and production quality of grazing yaks. Zhao et al. found that effective supplementary feeding can improve the production performance and breeding effect of calves [7] and lay a foundation for the later growth and development of calves. Xiccato et al. studied that the growth and development of yak calves can be promoted, and the production performance can be improved under the condition of supplementary feeding [8]. Mou Yongjuan found that the growth and development indexes of 3-month-old isolated weaned yak calves at 6 and 12 months of age were higher than those of fully suckling yak calves [9]. Cui Zhanhong found that by early weaning of yak calves and supplementing concentrate on the basis of feeding milk replacer and alfalfa hay [10], the digestibility and growth performance of yak calves were significantly improved, and the development of gastrointestinal tract and immune organs was promoted. Therefore, reasonable nutritional supplementation is essential for the early cultivation of yak calves.

Cade et al. found that secreted enzymes and enzymes in the pancreatic alkaline pancreatic juice promote digestion and absorption, islet hormones in the acinar maintain glucose homeostasis and regulate protein and fat metabolism [11]. The pancreas plays a vital role in regulating digestion and nutritional metabolism. Guo Long found that feeding a functional amino acid diet could regulate the development of dairy cow pancreas and improve its growth and production performance [12]. Yang et al. found that the concentration of insulin (INS) and insulin-like growth factor 1 (IGF-1) in plasma increased significantly when yaks were fed with a 6.90 MJ / kg high energy nutrition level [13], which could better regulate and improve the growth performance of yaks. Wu et al. affected the concentration of propionate in the rumen of yak calves by feeding starter and alfalfa hay before weaning [14], resulting in the upregulation and downregulation of cholecystokinin in the pancreatic secretion pathway, thereby increasing or reducing pancreatic α-amylase activity. Based on the above, nutritional supplementation of yak calves can effectively promote the pancreatic development of yak calves, but there are few reports on the supplementation of concentrate to regulate the pancreatic development of yak calves. The experiment explored the regulation of concentrate supplementation on the early growth and development quality of yak calves, clarified the effect of concentrate supplementation on pancreatic development and revealed the internal relationship between the overall development level of the pancreas and its physiological function. The purpose of this study was to improve the nutritional level of early diet by supplementing concentrate to yak calves, so as to promote the growth and development of calf body and pancreas and provide scientific reference for the research of early scientific cultivation technology of yak.

## 2. Materials and Methods

### 2.1. Experimental Animals and Group Design

Twenty one-month-old yak calves with healthy body condition and similar body weight were selected as experimental animals and randomly divided into two groups, five replicates in each group. The pre-feeding period of this experiment was 30 days and the trial period was 100 days. The experiment was carried out in Haibei Plateau Ecological Animal Husbandry Technology Demonstration Park in Haiyan County, Haibei Tibetan Autonomous Prefecture, Qinghai Province. The altitude of the experimental place was 3010 m, and the annual average temperature was 1.5 °C.

### 2.2. Experimental Diets and Feeding Management

At the beginning of the formal experiment, yak calves in the control group were fed milk replacer and alfalfa hay. The experimental group yak calves were fed milk replacer, alfalfa hay and concentrate. The two groups were fed the same amount of milk replacer every day. The alfalfa hay fed in the control group and the alfalfa hay and concentrate dry matter fed in the experimental group were the same, that is, the feeding amount of solid dry matter in the two groups was the same. The milk replacer powder and supplementary concentrate were purchased from Beijing Precision Animal Nutrition Research Center, alfalfa hay was purchased from local seasonal crops in Huangzhong district, Xining city, Qinghai Province. The nutritional levels of milk replacer, alfalfa hay and concentrate in the diet are shown in Table 1.

Yak calves were adapted to milk replacers in initial feeding trials. Milk replacer was fed three times per day, in the morning, middle of the day and evening. The milk replacer powder and boiled warm water were cooled to 42 °C and stirred at a ratio of 1:5, poured into the bottle and placed on the bottle rack, which was actively sucked by the calf. After 30 days of pre-feeding, yak calves gradually adapted to alfalfa hay and yak calves in the experimental group also adapted to alfalfa hay and concentrate. At the beginning of the formal feeding experiment, the feeding amount of milk replacer in both groups was 480 g/d, increasing by 12 g every 5 d. At the beginning of the experiment, the alfalfa hay in the control group was fed at 0.25 kg / day, and then increased by 57.5 g every 5 days. From day 1 to day 50, alfalfa hay and starter dry matter in the experimental group were fed at a ratio of 2:1. In the 51 ~ 100 days, according to the actual situation, the alfalfa hay and starter dry matter were adjusted by 1:1 ratio of feeding. The diet composition and nutritional level in the feeding experiment are shown in Table 2. During the experiment, single-column and single-circle feeding was carried out. Yaks could freely move inside and outside the bullpen, have enough space and sunlight and freely drink water and eat alfalfa hay and concentrate. Cleaning and disinfecting the bullpen were performed regularly every week.

### 2.3. Sample Collection

The slaughter test was performed after 12 h of fasting at the end of the feeding test. The abdominal cavity of yak calves was quickly opened, and tissue samples 1.5 cm long, 1.5 cm wide, and 2 cm high were collected from the pancreatic organs of yak calves. The tissue samples were collected, rinsed with prechilled saline and then quickly fixed in centrifuge tubes containing 4% paraformaldehyde. When collecting molecular samples of pancreas, the samples were trimmed into small square pieces of 0.5 cm, and each tube was loaded into a frozen tube with 2/3rd of the sample as quickly as possible, immediately put into a liquid nitrogen tank at −196 °C for refrigeration after completion and then transferred to an ultra-low temperature refrigerator at −80 °C for backup.

### 2.4. Measurement Indexes and Methods

#### 2.4.1. Growth Efficiency

In the feeding experiment, the yak calves were weighed every two weeks and the average daily gain was calculated. The body weight and dry matter intake of yak calves were recorded before slaughter and the average daily gain was calculated after the experiment.

#### 2.4.2. Dietary Nutrient Apparent Digestibility

Fecal samples and diet samples collected during the digestion test were dried to constant weight in an oven at 65 °C, moisture at room temperature, weighed and recorded and then crushed and passed through a 1 mm sieve to make the sample to be tested. The contents of CP, CF, NDF, ADF, Ca and P in diet samples and feces samples of each yak calf were determined and the apparent digestibility of nutrients was calculated.

#### 2.4.3. Morphological Development of the Pancreatic Tissue

Pancreatic tissue was collected and weighed after slaughter, and the organ index was calculated using the formula organ index (%) = organ weight / pre-slaughter weight × 100%. Pancreatic tissue-like specimens in 4% paraformaldehyde solution were fixed, flowed, dehydrated, waxed, embedded and then cut into 5 µm sections at room temperature and sealed after hematoxylin-eosin (HE) staining. The pancreatic tissue sections were observed under an inverted imaging microscope, and four randomly selected fields of view in each section were photographed. With a total area of 1×10^5^ um^2^ in the uniform field of view, the total area of the pancreas, the area of the exocrine part of the pancreas and the area of the endocrine part of the pancreas in their fields of view were measured and recorded using Image-Pro Plus 6.0 software. Calculation of total pancreatic exocrine part area share (%) = area of the exocrine part of the pancreas / total area of the pancreas × 100; total pancreatic endocrine part area share (%) = area of the endocrine part of the pancreas / total area of the pancreas × 100; area share of pancreatic endocrine to exocrine part (%) = area of the endocrine part of the pancreas / area of the exocrine part of the pancreas × 100.

#### 2.4.4. Functional Development of the Pancreas

Molecular samples of pancreatic tissue were collected and trimmed into small pieces in lyophilization tubes, which were rapidly frozen in liquid nitrogen for storage. The specimens were kept at room temperature after thawing. A certain amount of buffer (PBS) was added, the pancreatic samples of about 0.2 g were diluted to 10 times, and the specimens were homogenized well by hand or with a homogenizer. This was then centrifuged for about 20 min (2000–3000 rpm). The supernatant was carefully collected and dispensed. The absorbance (OD) of the supernatant was measured in the ELISA analyzer after following the steps of sample addition, enzyme addition, warming, liquid preparation, washing, color development and termination. After plotting the enzyme-linked immunosorbent assay (ELISA) kit results into a standard curve, the main digestive enzyme activities and hormone content in the pancreas were calculated.

#### 2.4.5. Metabolomics in the Pancreas

After grinding with pancreatic molecule-like liquid nitrogen, an 80% methanol solution was added and mixed well in an EP tube. Then, using a vortex shaker, it was given a good shake, set in a water bath for 5 min, and centrifuged at 4 °C for 20 min. After centrifugation, a certain volume of supernatant was diluted with mass spectrometry water until the methanol content was 53%. After centrifugation again for 20 min, a certain amount of supernatant was collected and sent into the Liquid Chromatography Mass Spectrometer (LC-MS) of a liquid chromatograph–mass spectrometer for analysis. Meta X 21.0 software was used to analyze the metabolites, and the variable importance in the projection (VIP) was obtained. SPSS 25.0 software was used to calculate the statistically significant *p*-value and fold change (FC) of each metabolite and finally, the differential metabolites were screened. When screening for differential metabolites, the thresholds were set to VIP > 1.0, FC > 1.2, or FC < 0.833, and *p*-value < 0.05.

### 2.5. Processing of Data

The growth performance indexes, apparent digestibility of dietary nutrients, morphological development indexes of pancreatic tissue, functional development indexes and metabolomics data of yak calves were recorded. Excel 2010 was used to organize the data, and a statistical t-test was performed using SPSS 25.0 software. *P*-value ≤ 0.05 was used as the criterion to determine the significance of differences, with *p*-value < 0.05 indicating that there were significant differences and *p*-value > 0.05 indicating that there were no significant differences.

## 3. Results

### 3.1. Effects of Concentrate Supplementation on Growth Performance of Yak Calves

The dry matter intake and body weight of yak calves in the test group were significantly higher than those in the control group (*p* < 0.05) (Table 3).

The apparent nutrient digestibility of crude protein (CP), crude fat (EE), calcium (Ca) and phosphorus (P) in the test group rations was significantly higher than that of the control group (*p* < 0.05), while the apparent nutrient digestibility of neutral detergent fiber NDF and acid detergent fiber ADF in the test group was significantly lower than that of the control group (*p* < 0.05) (Table 4).

### 3.2. Effects of Concentrate Supplementation on Morphological Development of Pancreatic Tissue in Yak Calves

The pancreatic weight of yak calves and their organ index were significantly higher in the test group than in the control group (*p* < 0.05) (Table 5).

The pancreas of yak calves consists of two parts: the endocrine and exocrine parts, namely the islets and the alveoli (Figure 1A). The exocrine part of the pancreas mainly consists of the alveoli, intercalary ducts, interlobular ducts, and interlobular connective tissue (Figure 1B). The intercalary duct has a small lumen and is connected to the inter-lobular duct at the end. The lumen of the interlobular duct is larger and forms interlobular connective tissue after exiting the interlobular lobules of the pancreas.

Figure 2A–E of the control group showed sparse and scattered distribution of pancreatic tissue structures, more interlobular ducts and blood vessels and less composition of internal and external secretory parts. Figure 2F–J of the test group showed dense internal pancreatic tissue arrangement with fewer ducts and more distribution of alveoli and islets relative to the control group. In the test group, the endocrine and exocrine parts of the pancreas were more closely arranged, with more widely distributed alveolar cells and islet cells relative to the control group.

The total area ratio of the exocrine part of the pancreas and the total area ratio of the endocrine part of the pancreas in the test group of yak calves were significantly higher than those in the control group (*p* < 0.05), while the area ratio of the exocrine part of the pancreas in the test group was significantly lower than that in the control group (*p* < 0.05) (Table 6).

### 3.3. Effects of Concentrate Supplementation on Pancreatic Function Development in Yak Calves

The activities of pancreatic amylase, pancreatic lipase, pancreatic protease and chymotrypsin in the test group of yak calves were significantly higher than those in the control group (*p* < 0.05) (Table 7).

The levels of glucagon, insulin and pancreatic polypeptide in the test group of yak calves were significantly higher than those in the control group (*p* < 0.05) (Table 8).

### 3.4. Effects of Concentrate Supplementation on Pancreatic Metabolomics in Yak Calves

In the positive ion mode (Table 9), the differential metabolites 2-acetamido-3-(4-methoxyphenyl)propionic acid, D-(+)-proline, hypoxanthine, ACar (7:0), O-acetyl-L-carnitine, ellagic acid, ACar (22:5), adipamide, acetylcysteine, corey lactone diol UDP, gamma-glutamylcysteine, Prostaglandin A3, uric acid, 2-oxa-4-azatetracyclic [6.3.1.1~6,10~.0~1,5~] tridecan-3-one, creatinine, tetrahydrocorticosterone, thiazoli-dine-4-carboxylic acid, LSD-d3, pyridoxamine, nicotinamide, 23-deoxycholic acid, MJN110, xanthine, 5-oxoproline, 2-phenylacetamide, tropolone, D-serine, pipecolic acid, PC (17:2/18:5), gamma-glutamyl-leucine, O-phosphocholine, N2-tetrahydrofuran-2-ylmethyl-4-(4-fluorophenyl)-1,3-thiazol-2-amine, ACar (18:2), uracil 1-beta-D-arabinofuranoside, asparagine, DL-tryptophan, indole-3-propenoic acid, ribulose-5- phosphate, temazepam, 5-methyl-5-thioadenosine, (+/-)5(6)-EET, 2-hydroxycinnamic acid, L-phenylalanine, 6-methylquinoline, thymine and trehalose were significantly higher than the control (*p* < 0.05). In contrast, the levels of dopaquinone, phenylpyruvic acid, PC (20:5e/4:0), PE (18:5e/6:0), PE (18:1/20:5), PC (18:4e/6:0), PC (17:0/17:1) and PC (16:0e/4:0) in the test group were significantly lower than those in the control group (*p* < 0.05).

In the negative ion mode (Table 10), the differential metabolites LPE (20:5), Gamma-Glutamylcysteine, docosapentaenoic acid, PC (16:1e/18:2), LPG (22:5), FAHFA (18:1/20:3), PC (14:0/18:2), PE (18:3e/18:2), PE (18:2/18:3), FAHFA (18:2/20:4), LPG (22:4), OxPC (18:1-18:2+2O), octanedioic acid, D-proline, cis-5,8,11,14,17-eicosapentaenoic acid, adrenaline, docosahexaenoic acid, xanthosine and LPS (22:4) were significantly higher (*p* < 0.05) than those of the control group. And the content of 3-(methylthio)-5H-[1,2,4]triazino[5,6-b]indole in the test group was significantly lower than that in the control group (*p* < 0.05).

## 4. Discussion

### 4.1. Effects of Concentrate Supplementation on Growth Performance of Yak Calves

The results showed that supplementary concentrate could increase the dry matter intake, body weight and average daily gain of yak calves. Supplemental feeding of concentrates by herders during the green and dry grass periods can effectively increase the body weight of yak calves and facilitate their early growth and development [15]. The average daily gain (ADG) and slaughter weight of beef cattle were significantly increased after supplementation with concentrates such as corn meal during fattening [16]. It is indicated that the appropriate supplemental concentrate can promote the development and growth of muscle carcasses of yak calves, and significantly affect and improve the growth and development performance of yak calves. It is concluded that supplementary concentrate plays an important role in promoting the growth and development of yak calves.

Meanwhile, supplementary concentrate promoted the digestion and absorption of protein, fat, calcium and phosphorus, and inhibited the utilization of fiber in yak calves, which corresponded to the increase of solid dry matter mass and fully verified that the test results were more reasonable. Supplemental feeding of concentrates to improve calves significantly improved the digestibility of crude protein (CP) and crude fat (EE) [17]. Suckling yaks could meet their energy and protein requirements after supplementing with appropriate amounts of concentrate [18]. In particular, supplementary concentrate greatly improved the digestion and utilization of protein and fat in yak calves, enhanced feed utilization efficiency, promoted the early growth and development of yak calves and also laid a good foundation for high-quality production of yak calves in the later stage.

### 4.2. Effects of Concentrate Supplementation on Pancreatic Development of Yak Calves

Supplemental feeding of concentrate can have a significant effect on the pancreatic weight and pancreatic organ index of yak calves. Appropriate increases in concentrate types of feeds along with the increase of pancreatic secretion and release of insulin from the pancreatic islets of livestock can facilitate the growth and development of the pancreatic gland itself [19]. Supplementing concentrate also reduced the distribution of intercalated ducts and ducts in the pancreas of yak calves, increased the distribution density of pancreatic acinar and islets and increased the area and proportion of endocrine and exocrine parts of the pancreas. Svensson et al. compared the capillary area in the endocrine and exocrine parenchyma of the pancreas and found that the capillary volume in the islets was about 3.5% [20], while the volume of capillaries in the exocrine pancreas was significantly lower at 2%. The reason for the discrepancy between these parameters may be the lack of lymphatic capillaries and the relatively small capillary lumen in the pancreatic islets. In particular, there was a significant increase in both the total percentage of area of the exocrine part of the pancreas and the total percentage of area of the endocrine part. The density of islet cells is higher in parts of the pancreas than in the alveolar region [21], and the number of cells is higher, especially in islets rich in glucagon. The protein content of dairy cows that were given supplemental concentrates affected the exocrine function of the pancreas [22]. The area ratio of the exocrine part of the pancreas was significantly lower because the supplemented concentrate increased the area of the exocrine part of the pancreas, thus making the area ratio of the endocrine part of the pancreas relatively low, which also reflected that the supplemented concentrate had the greatest effect on the exocrine part of the pancreas. This indicates that supplemental concentrate feeding has a positive effect on the internal alveoli and islets, endocrine and exocrine parts of the pancreas of suckling yak calves and also has a positive effect on the growth and development of the pancreas of yak calves.

This study found that supplementary concentrate can increase the activity of the main digestive enzymes and hormone content in the pancreas of yak calves, especially the activity of pancreatic amylase, pancreatic lipase, pancreatic protease and chymotrypsin and the content of glucagon, insulin and pancreatic polypeptide at significant levels. Supplemental feeding can significantly increase the activity of duodenal pancreatic amylase [23], pancreatic lipase, pancreatic protease and chymotrypsin in grazing lambs to digest and absorb nutrients in the diet through a series of biochemical reactions and promote the growth, development and production quality of lambs. Since the alveoli of the pancreas secrete the major digestive enzymes such as pancreatic amylase, pancreatic lipase, pancreatic protease and chymotrypsin [24], Islet A, B, D and PP cells secrete mainly the hormones glucagon, insulin, growth inhibitory hormone and pancreatic polypeptide [25]. Side by side, it was shown that supplementation with concentrate increased the alveoli and islet cells of the pancreas of suckling yak calves and also relatively increased the density of distribution of their endocrine and exocrine divisions. This is in agreement with the above results and discussion and serves as a good illustration. Thus, it was concluded that supplementation with concentrate not only increased the activity of major digestive enzymes and hormone content of the pancreas of suckling yak calves, but also promoted the digestion and absorption of nutrients in the diet and improved the growth and development of the pancreas of yak calves.

### 4.3. Effects of Concentrate Supplementation on Pancreatic Metabolomics in Yak Calves

Supplemental feeding of concentrates increased the levels of differential metabolites associated with resistance to oxidative stress in the pancreas of suckling yak calves. D-proline could induce sedation and hypnosis through N-methyl-D-aspartate glutamate receptors and glycine receptors [26] and could effectively inhibit stress behavior in response to isolation-induced stress. Mammals improve stress tolerance through non-cyclic utilization of purine metabolites, and increased hypoxanthine levels can improve reuse during cyclic metabolism [27], thus also achieving a reduction in oxidative stress. Supplementation with concentrate increased the levels of the differential metabolites D-proline and hypoxanthine in the pancreas of yak calves in the trial, which effectively reduced the stress behavior of suckling yak calves and facilitated their normal and effective daily feeding and exercise. Acetylcysteine is an acetylated precursor of cysteine, which is involved in the synthesis of glutathione in the body and has antioxidant, anti-inflammatory, antibacterial and fat metabolism regulating effects. Acetylcysteine has positive effects on maintaining the intestinal health of livestock [28], alleviating oxidative stress and improving their reproductive performance. Supplemental feeding of concentrate increased the acetylcysteine content in suckling yak calves, promoting their good development and healthy growth. Supplemental feeding of concentrates increased gamma-glutamylcysteine levels in the pancreas of suckling yak calves, as found in the trial. In contrast, gamma-glutamylcysteine is abundant in mammals and is involved in the synthesis of glutathione in the presence of its ligase [29], preventing potential adverse reactions associated with reactive oxygen species and associated redox reactions that may induce oxidative stress and may be linked to innate detoxification processes, ensuring, to some extent, healthy organism growth. Thiazolidine-4-carboxylic acid was modified by bioactivity and inhibited apoptosis caused by oxidative stress by enhancing the activity of catalase and improving the ability to bind to proteins, thereby regulating bioactivity [30]. The increased thiazolidine-4-carboxylic acid content in the pancreas effectively alleviated the stressful behavior of yak calves, creating favorable conditions for their healthy growth and development. Increased serum levels of piperidine acid enhanced the ability of mammalian cells to resist oxidative stress and also hindered DNA damage and cell growth arrest induced by oxidative stress [31]. Piperidinic acid, a non-protein amino acid derived from the catabolic metabolism of lysine [32], is an important modulator of body immunity and piperidinic acid levels increase upon infection with pathogens and by increasing free radical levels with the acquisition of systemic resistance. Supplemental feeding of concentrate increased piperidinic acid levels in the pancreas of yak calves, alleviated cell growth stagnation due to oxidative stress, improved early body immunity and promoted healthy growth and development later in life.

Supplemental feeding of concentrates increased the levels of differential metabolites in the pancreas of suckling yak calves associated with the promotion of their growth and development. Supplemental ellagic acid feeding to suckling foals increased plasma protein concentration and body weight size [33], improved nutrient metabolism, further regulated their organism health and promoted good growth and development of suckling foals. In this experiment, supplemental feeding of concentrate increased the content of the differential metabolite ellagic acid in the pancreas of yak calves, which was beneficial in promoting the productive performance and growth of suckling yak calves. Nicotinamide promotes pancreatic progenitor cell differentiation through its off-target effect as a kinase inhibitor [34], while the development of pancreatic progenitor cells and endocrine cells can be induced. By supplementing concentrate to yak calves, the content of nicotinamide in the pancreas increased, which actively regulated the differentiation of pancreatic cells and promoted the good development of the pancreas. Tropolone visualizes the location of genes and cells in vivo based on a dual-isotope SPECT study [35], and glucagon and insulin positive cells can be observed by immunohistochemistry. Supplemental feeding of concentrate in this experiment increased the level of tropolone in the pancreas of suckling yak calves, corresponding to the previous increase in glucagon and insulin, which is consistent with and validates the previous study. The administration of D-serine to rats improved the positive and negative symptoms and their resulting cognitive dysfunction after schizophrenia [36]. The increased level of D-serine in the pancreas of yak calves after supplementation with concentrate in the experiment improved the learning and cognitive function of suckling yak calves. Ribulose-5-phosphate is a very important anabolic intermediate in the pentose phosphate pathway [37], inducing redox metabolism, synthetic glucose metabolism and fatty acid synthesis, and playing a regulatory role in cellular energy metabolism. Supplemental feeding of concentrate in the experiment increased the content of ribulose-5-phosphate in the pancreas of suckling yak calves, indicating that supplemental feeding of concentrate can improve the regulation of glucose metabolism and lipid metabolism in early yak calves and create favorable conditions for their healthy growth. (+/-)5(6)-EET has been shown to have biological effects such as vasodilatory, anti-inflammatory and sodium-urea effects, as well as to stimulate insulin and glucagon secretion from an isolated pancreas [38], increase insulin sensitivity and lower blood pressure in rats, and increasing (+/-)5(6)-EET levels improved insulin sensitivity and reduced diabetes-related cardiovascular and renal damage. In this experiment, supplementation with concentrate increased (+/-)5(6)-EET levels in the pancreas of suckling yak calves, verified increased insulin and glucagon levels in the pancreas and positively regulated insulin sensitivity and glucose levels during the growth and development of early yak calves. It was found that 2-hydroxycinnamic acid inhibited oleic acid-induced lipid accumulation in HepG2 cells [39], decreased TC and TG content in cells, enhanced glucose utilization in cells and decreased sterol regulatory elements. The increased levels of 2-hydroxycinnamic acid in the pancreas suggest that supplementation with concentrate can modulate the effects of cellular glucolipid metabolism in suckling yak calves, improve lipid accumulation in cells and increase glucose utilization. Increasing the content of L-phenylalanine promoted the secretion of pancreatic amylase and improved the exocrine capacity of pancreatic alveolar cells by increasing the expression of pancreatic amylase mRNA and forming a protein translation initiation complex [12]. Supplemental feeding of concentrate in the experiment increased the L-phenylalanine content in the pancreas of suckling yak calves, indicating that supplemental feeding of concentrate can promote the secretion of digestive enzymes from early yak calves’ glandular alveolar cells and the growth and development of the pancreas, as well as improving the digestion and absorption of feed in the small intestine, and increasing its utilization efficiency.

Supplemental feeding of concentrates increased the content of differential metabolites associated with anti-inflammatory preventive diseases in the pancreas of suckling yak calves. The prevalence of diabetes, acromegaly, and pancreatic inflammation were predicted using insulin activity and the ratio of pancreatic lipase concentration to creatinine [40]. The activity of pancreatic lipase and the content of the differential metabolite creatinine in the pancreas are crucial for the health of the yak calf organism, and, to some extent, they reflect the good growth and development status. Glucocorticoids containing tetrahydrocorticosterone can effectively regulate body homeostasis and glucose homeostasis [41], where excess glucocorticoids can trigger insulin resistance and type 2 diabetes mellitus. In the present experiment, supplementation with concentrate significantly increased the level of tetrahydrocorticosterone in the pancreas of yak calves, suggesting that it could be used as a potential marker for pancreatic-related metabolic diseases. Pyridoxamine could reduce the accumulation of glycosylation end products in the brain microvascular wall and improve the ischemic and hypoxic state of the type 2 diabetic brain [42], resulting in a significant reduction in the latency of diabetes. The significant increase in pyridoxamine content in the pancreas of yak calves in this experiment suggests that supplementation with concentrates can prevent potential type 2 diabetes and improve different degrees of behavioral and cognitive dysfunction to some extent. Pancreatic β-cells play an important role in glucose-dependent insulin secretion, and hyper-glycemic glucotoxicity causes pancreatic β-cell dysfunction and decreases insulin secretion [43]. KUP-1, a xanthine derivative, inhibited the induction of hyperglycemic glucotoxicity, thereby stabilizing β-cell function and increasing insulin secretion. Supplementation with concentrate increased the xanthine content in the pancreas of yak calves, indicating that supplementation with concentrate can effectively control the metabolic syndrome-related diseases induced by hyperglycemia. It was determined that 5-oxoproline could be identified as a candidate diagnostic marker to distinguish mucinous from non-mucinous pancreatic cysts [44], and its establishment could allow early detection and accurate diagnosis of pancreatic cysts and provide guidance for clinical treatment policy. Compounds synthesized from asparagine conversion exhibited good biological activity in terms of anti-inflammatory, hypotensive and hemostatic properties [45]. Supplemental feeding of concentrate increased the asparagine content in the pancreas of suckling yak calves, which improved the anti-inflammatory and hemostatic ability of early yak calves. Pro-inflammatory factors in wound inflammation induce the activity of the indoleamine 2,3-dioxygenase IDO1 [46], and DL-tryptophan is an effective inhibitor of IDO1, thereby accelerating the rate of wound healing. The increase in pancreatic DL-tryptophan content in the experiment suggests that supplemental concentrate feeding improves the ability of suckling yak calves to suppress inflammatory responses and improve self-healing. Indole-3-acrylic acid, a tryptophan metabolite produced by the bacterial species Pepto-coccosis [47], is a powerful activator of antioxidant mechanisms in cells, promotes intestinal barrier function and decreases inflammation. The increase in indole-3-acrylic acid content in the pancreas suggests that supplementation with concentrate improves the cellular antioxidant capacity of suckling yak calves, reduces the intestinal inflammatory response and promotes healthy growth. Thymine derivatives had strong antioxidant and antibacterial properties [48], exhibited excellent hematological and biocompatibility and treated wounds with fewer inflammatory cells and faster epithelial tissue regeneration and neointima formation. The increase in pancreatic thymidine content in the trial suggests that supplemental concentrate feeding improved anti-inflammatory and tissue re-generation in suckling yak calves and facilitated early thriving of yak calves. Trehalose ameliorated the abnormal autophagy protein expression caused by the pancreas in diabetic mice by activating the AKT/GSK3β signaling pathway [49] and increasing the level of autophagy, thus, having a protective effect on the diabetic pancreas. Trehalose reduced the abnormal expression of apoptotic and pyrogenic proteins in the pancreas of diabetic mice and improved the pancreatic damage caused by diabetes mellitus [50]. The increase in pancreatic trehalose content in the experiment suggests that supplemental concentrate feeding can protect the pancreas of suckling yak calves from damage caused by diabetes and is beneficial to their good pancreatic development and the healthy growth of the organism. Docosahexaenoic acid is a key regulator of cardiomyocyte membranes [51], responsible for maintaining cholesterol homeostasis and has a direct impact on eicosanoid metabolism. Docosapentaenoic acid and docosahexaenoic acid, as bioactive lipid mediators, have potent anti-inflammatory and immunomodulatory effects in vitro and in vivo. Supplemental feeding of concentrates increased docosapentaenoic acid and docosahexaenoic acid levels in the pancreas of yak calves, indicating that supplemental feeding of concentrates can improve the immunity of suckling yak calves and effectively promote their healthy growth. Branched-chain fatty acid esters of hydroxy fatty acids (FAHFAs) [52], a product of endogenous synthesis in mammalian tissues, reduced the production of pro-inflammatory cytokines through adipose tissue macrophages and insulin resistance in obese mice. Supplementation of concentrate in the experiment increased the content of FAHFA (18:1/20:3) and FAHFA (18:2/20:4) in the pancreas of yak calves, indicating that supplementation of concentrate enhanced the anti-inflammatory and anti-diabetic properties of early yak calves. In vitro, adrenic acid increases macrophage phagocytosis and cytotoxicity [53], clears neutrophils from peritonitis in mice and has the potential to prevent and resist inflammation. Supplemental feeding of concentrate in the experiment increased the level of adrenic acid in the pancreas of suckling yak calves, suggesting that supplemental feeding of concentrate can improve the anti-inflammatory and antibacterial capacity of early yak calves. Xanthosine could inhibit inflammatory signaling pathways [54] antimicrobial gene upregulation and cell adhesion molecules and induce downregulation of inflammatory signals and up-regulation of antimicrobial genes. Supplemental feeding of concentrate increased the content of xanthosine in the pancreas in the experiment, suggesting that supplemental feeding of concentrate can improve the anti-inflammatory and antibacterial ability of suckling yak calves and have a positive effect on their growth and development and production performance.

## 5. Conclusions

In summary, it is suggested that supplementary feeding concentrate promoted the good development of morphological and functional characteristics of pancreas in early weaned yak calves, thus improving their digestion and utilization efficiency of feed nutrients and increasing the activity of main digestive enzymes and hormone levels in the pancreas of yak calves. This had positive regulatory effects on the endocrine and exocrine parts of the pancreas, especially in the increase in the content of small molecule differential metabolites so as to achieve the effects of relieving early oxidative stress, enhancing the anti-inflammatory and antibacterial ability to prevent diseases, which will provide important references to the healthy and high-quality rearing of early weaned yak calves.

## Figures and Tables

**Figure 1 animals-12-02563-f001:**
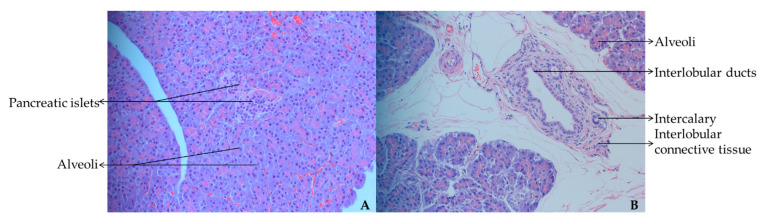
(**A**) Internal morphological structure of pancreatic tissue under 10× microscope; (**B**) Internal morphological structure of pancreatic tissue under 20× microscope.

**Figure 2 animals-12-02563-f002:**
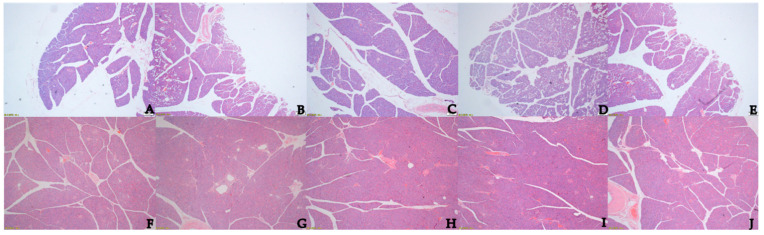
(**A**–**E**) Effect of no supplementation of concentrate on morphological development of pancreatic tissue in yak calves under 10× microscope; (**F**–**J**) Effect of concentrate supplementation on morphological development of pancreatic tissue in yak calves under 10× microscope.

**Table 1 animals-12-02563-t001:** Nutritional components of milk replacer, alfalfa hay and supplementary concentrate (dry matter basis) (Unit: %).

Items	Milk Replacer Powder	Alfalfa Hay	Concentrates
Crude protein (CP)	26.24	12.50	20.00
Ether extract (EE)	27.90	0.90	4.70
Neutral detergent fiber (NDF)	-	56.45	10.90
Acid detergent fiber (ADF)	-	40.40	4.10
Calcium (Ca)	2.50	0.98	0.80
Phosphorus (P)	1.40	0.18	0.45

**Table 2 animals-12-02563-t002:** Composition and nutrient levels of diets (dry matter basis) (Unit: %).

Items	Days 1 to 50		Days 51 to 100	
	Control Group	Test Group	Control Group	Test Group
Ingredients				
Milk replacer	51.21	51.21	37.63	37.63
Alfalfa hay	48.79	32.53	62.37	31.185
Supplemental concentrate	-	16.26	-	31.185
Total	100.00	100.00	100.00	100.00
Nutrient levels				
Crude protein (CP)	19.54	20.76	17.67	20.01
Ether extract (EE)	14.73	15.34	11.06	12.25
Neutral detergent fiber (NDF)	27.54	20.14	35.21	21.00
Acid detergent fiber (ADF)	19.71	13.81	25.20	13.88
Calcium (Ca)	1.76	1.73	1.55	1.50
Phosphorus (P)	0.80	0.85	0.64	0.72

**Table 3 animals-12-02563-t003:** Effect of concentrate supplementation on growth performance of yak calves.

Items	Control Group	Test Group	*p*-Value
Dry matter intake (g)	1 336.37 ± 94.85 ^b^	1 469.97 ± 6.24 ^a^	0.003
Body weight (kg)	69.30 ± 1.16 ^b^	76.13 ± 1.80 ^a^	0.005
Average daily gain (g)	426.68 ± 8.89	449.23 ± 10.65	0.121

^a,b^ within a row with different superscripts means significantly difference. Different letters on the shoulder of the same row of data indicate significant differences (*p* < 0.05), and no letters or the same letters indicate non-significant differences (*p* > 0.05). The same as the table below.

**Table 4 animals-12-02563-t004:** Effect of concentrate supplementation on nutrient apparent digestibility of yak calves (Unit: %).

Items	Control Group	Test Group	*p*-Value
CP	78.39 ± 0.82 ^b^	82.50 ± 1.38 ^a^	0.028
EE	89.83 ± 0.57 ^b^	92.30 ± 0.91 ^a^	0.044
NDF	81.90 ± 1.04 ^b^	73.79 ± 3.41 ^a^	0.046
ADF	73.63 ± 1.27 ^b^	69.79 ± 0.36 ^a^	0.016
Ca	62.76 ± 2.91 ^b^	75.26 ± 1.29 ^a^	<0.001
P	26.86 ± 1.73 ^b^	29.74 ± 2.63 ^a^	<0.001

^a,b^ within a row with different superscripts means significantly difference. Different letters on the shoulder of the same row of data indicate significant differences (*p* < 0.05), and no letters or the same letters indicate non-significant differences (*p* > 0.05). The same as the table below.

**Table 5 animals-12-02563-t005:** Effect of concentrate supplementation on pancreatic organ index of yak calves.

Items	Control Group	Test Group	*p*-Value
Pancreas weight (g)	67.86 ± 1.62 ^b^	86.58 ± 2.48 ^a^	0.048
Organ index (%)	0.10 ± 0.03 ^b^	0.11 ± 0.02 ^a^	0.004

^a,b^ within a row with different superscripts means significantly difference. Different letters on the shoulder of the same row of data indicate significant differences (*p* < 0.05), and no letters or the same letters indicate non-significant differences (*p* > 0.05). The same as the table below.

**Table 6 animals-12-02563-t006:** Effect of concentrate supplementation on the area of internal and external secretion of pancreas in yak calves (Unit: %).

Items	Control Group	Test Group	*p*-Value
Total pancreatic exocrine part area share	0.8275 ± 0.0272 ^b^	0.9039 ± 0.0197 ^a^	0.029
Total pancreatic endocrine part area share	0.0307 ± 0.0024 ^b^	0.0390 ± 0.0032 ^a^	0.044
Area share of pancreatic endocrine to exocrine part	0.0384 ± 0.0012 ^b^	0.0353 ± 0.0007 ^a^	0.030

^a,b^ within a row with different superscripts means significantly difference. Different letters on the shoulder of the same row of data indicate significant differences (*p* < 0.05), and no letters or the same letters indicate non-significant differences (*p* > 0.05). The same as the table below.

**Table 7 animals-12-02563-t007:** Effect of Supplemental Concentrate on Main Digestive Enzyme Activities in Pancreas of Yak Calves (Unit: IU/L).

Items	Control Group	Test Group	*p*-Value
Pancreatic amylase	41.746 ± 0.127 ^b^	45.434 ± 0.249 ^a^	0.039
Pancreatic lipase	140.407 ± 4.020 ^b^	155.161 ± 3.464 ^a^	0.009
Pancreatic protease	175.410 ± 1.844 ^b^	196.966 ± 2.888 ^a^	0.005
Chymotrypsin	195.910 ± 6.583 ^b^	224.350 ± 5.938 ^a^	0.002

^a,b^ within a row with different superscripts means significantly difference. Different letters on the shoulder of the same row of data indicate significant differences (*p* < 0.05), and no letters or the same letters indicate non-significant differences (*p* > 0.05). The same as the table below.

**Table 8 animals-12-02563-t008:** Effect of concentrate supplementation on pancreatic hormone content in yak calves (Unit: Ug/L).

Items	Control Group	Test Group	*p*-Value
Glucagon	0.281 ± 0.004 ^b^	0.312 ± 0.002 ^a^	0.042
Insulin	1.044 ± 0.034 ^b^	1.135 ± 0.011 ^a^	0.012
Glucagon	2.744 ± 0.040 ^b^	3.087 ± 0.050 ^a^	0.041
Growth inhibitors	0.293 ± 0.003	0.299 ± 0.002	0.587

^a,b^ within a row with different superscripts means significantly difference. Different letters on the shoulder of the same row of data indicate significant differences (*p* < 0.05), and no letters or the same letters indicate non-significant differences (*p* > 0.05).

**Table 9 animals-12-02563-t009:** Differential metabolites in positive ion mode.

Items	FC	VIP	*p*-Value	Trend
Dopaquinone	0.2790	2.4151	0.0022	↓ ^1^
2-acetamido-3-(4-methoxyphenyl)propanoic acid	2.4534	1.9560	0.0033	↑
D-(+)-Proline	2.0918	1.9263	0.0070	↑
Hypoxanthine	1.8473	1.6555	0.0074	↑
ACar ^2^ (7:0)	2.1751	1.3468	0.0077	↑
O-Acetyl-L-carnitine	2.3309	2.0457	0.0088	↑
Ellagic acid	1.7811	1.8492	0.0089	↑
ACar (22:5)	2.9093	1.4077	0.0097	↑
Adipamide	1.7423	1.7909	0.0105	↑
Acetylcysteine	1.9873	1.7905	0.0107	↑
Phenylpyruvic acid	0.5255	1.2840	0.0112	↓
Corey Lactone Diol	1.7025	1.0985	0.0115	↑
UDP ^3^	2.0778	1.3672	0.0116	↑
PC ^4^ (20:5e/4:0)	0.1133	2.4254	0.0117	↓
Gamma-Glutamylcysteine	2.1049	1.3942	0.0130	↑
Prostaglandin A3	1.5521	1.3625	0.0159	↑
Uric acid	1.3932	1.0666	0.0177	↑
2-oxa-4-azatetracyclo [6.3.1.1~6,10~.0~1,5~] tridecan-3-one	1.8355	1.3543	0.0182	↑
Creatinine	1.6563	1.3009	0.0200	↑
Tetrahydrocorticosterone	2.6623	1.1505	0.0262	↑
Thiazolidine-4-carboxylic acid	2.0473	1.2061	0.0266	↑
LSD ^5^-d3	1.8016	1.7897	0.0270	↑
PE ^6^ (18:5e/6:0)	0.1866	2.1564	0.0274	↓
Pyridoxamine	1.8983	1.5306	0.0281	↑
Nicotinamide	1.5085	1.1741	0.0281	↑
PE (18:1/20:5)	0.6259	1.9053	0.0284	↓
23-Nordeoxycholic acid	1.7533	1.1172	0.0285	↑
MJN110 ^7^	2.0824	1.4017	0.0310	↑
Xanthine	1.5591	1.1207	0.0329	↑
5-oxoproline	1.5393	1.1173	0.0337	↑
2-Phenylacetamide	1.7269	1.1676	0.0340	↑
Tropolone	1.7080	1.0937	0.0341	↑
D-Serine	1.6387	1.1970	0.0347	↑
Pipecolic acid	1.6846	1.2970	0.0347	↑
PC (17:2/18:5)	1.9933	1.4874	0.0364	↑
PC (18:4e/6:0)	0.1144	2.2393	0.0382	↓
Gamma-Glutamylleucine	1.8133	1.1139	0.0384	↑
O-Phosphocolamine	1.4013	1.1532	0.0384	↑
N2-tetrahydrofuran-2-ylmethyl-4-(4-fluorophenyl)-1,3-thiazol-2-amine	2.4647	1.2867	0.0386	↑
ACar (18:2)	2.4467	1.1459	0.0388	↑
Uracil 1-beta-D-arabinofuranoside	1.7342	1.7156	0.0393	↑
Asparagine	1.5276	1.0634	0.0397	↑
PC (17:0/17:1)	0.6580	1.5837	0.0405	↓
DL-Tryptophan	1.6126	1.1470	0.0417	↑
Indole-3-acrylic acid	1.6111	1.1445	0.0422	↑
Ribulose-5-phosphate	1.7376	1.1642	0.0440	↑
Temazepam	1.7146	1.3424	0.0441	↑
5'-S-Methyl-5'-thioadenosine	1.7697	1.2447	0.0443	↑
(+/-)5(6)-EET ^8^	1.8939	1.3987	0.0445	↑
2-Hydroxycinnamic acid	1.6544	1.1037	0.0453	↑
L-Phenylalanine	1.5400	1.1056	0.0454	↑
6-Methylquinoline	1.5664	1.0629	0.0461	↑
Thymine	1.7435	1.0853	0.0484	↑
Trehalose	1.6726	1.2917	0.0493	↑
PC (16:0e/4:0)	0.6476	1.2950	0.0494	↓

^1^ ↑ represents upward adjustment; ↓ represents downward adjustment. ^2^ ACar represents acylcarnitine. ^3^ UDP represents uridine diphosphate. ^4^ PC represents phosphatidycholine. ^5^ LSD represents lysergic acid diethylamide. ^6^ PE represents phosphatidylethanolamine. ^7^ MJN110 represents monoacylglycerol lipase inhibitor. ^8^ EET represents epoxyeicosatrienoic acid. The same as the table below.

**Table 10 animals-12-02563-t010:** Differential metabolites in negative ion mode.

Items	FC	VIP	*p*-Value	Trend
LPE ^1^ (20:5)	2.8219	2.4732	0.0008	↑
Gamma-Glutamylcysteine	2.9828	1.5129	0.0074	↑
Docosapentaenoic acid	2.0154	2.0502	0.0119	↑
PC (16:1e/18:2)	2.8861	2.0992	0.0140	↑
LPG ^2^ (22:5)	2.8985	1.4841	0.0149	↑
FAHFA ^3^ (18:1/20:3)	1.6771	1.9790	0.0242	↑
PC (14:0/18:2)	3.5938	1.0535	0.0274	↑
PE (18:3e/18:2)	2.8885	1.2124	0.0291	↑
PE (18:2/18:3)	2.5257	1.1019	0.0296	↑
FAHFA (18:2/20:4)	1.4077	1.2318	0.0307	↑
LPG (22:4)	2.8138	1.5457	0.0320	↑
OxPC ^4^ (18:1-18:2+2O)	3.1536	1.1076	0.0352	↑
Octanedioic acid	1.3331	1.2553	0.0357	↑
*D*-Proline	2.5356	1.1978	0.0370	↑
*Cis*-5,8,11,14,17-Eicosapentaenoic acid	1.8382	1.7902	0.0411	↑
Adrenic acid	1.9676	1.2603	0.0414	↑
3-(methylsulfanyl)-5H-[1,2,4]triazino [5,6-b]indole	0.2923	3.2189	0.0429	↓
Docosahexaenoic acid	1.4759	1.5493	0.0474	↑
Xanthosine	2.2922	1.0191	0.0491	↑
LPS ^5^ (22:4)	2.2800	2.2501	0.0493	↑

^1^ LPE represents lyso-phosphatidylethanolamine. ^2^ LPG represents lyso-phosphatidylglycerol. ^3^ FAHFA represents fatty acid esters of hydroxy fatty acids. ^4^ OxPC represents oxidized Phosphatidylcholines. ^5^ LPS represents lyso-phosphatidylserine.

## Data Availability

All image table data from the trial can be found in the article. The raw data involved in metabolomics are publicly available and can be found in MTBLS5502.

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
