# Peer review of "The Effect of Supplemental Concentrate Feeding on the Morphological and Functional Development of the Pancreas in Early Weaned Yak Calves"

_animals, 2022, doi:10.3390/ani12192563_

Round 1

Reviewer 1 Report

The authors investigated the effects of supplementing feed with concentrate on the development of the pancreas in suckling yak calves. The subject of this study is suitable for the “Animals” journal. The authors indicated that supplemental concentrate feeding during lactation could exploit the growth potential and improve the digestive utilization of nutrients in the diet of yak calves and actively promote the positive development of calf pancreas morphology and function. This is an exciting study, but due to the concerns I have stated below, the manuscript is unsuitable for publication in Animals Journal.

1. There is a lot of redundant information in the introduction, such as the general functions of the pancreas and the effect of taurine supplementation on production performance and pancreatic development in the broiler.

2. The aim of the study is not sufficiently explained in the introduction.

3. The feeding of yak calves in the control group and test group after weaning and the feeding during the experiment were not written clearly enough in the materials and methods section. It's confusing!!

4. Control and test groups should be fed at the same level regarding nutrient content so that the expected effects can be evaluated. I understand that the test group was given concentrated feed in addition to the control group, which indicates that the test group is getting more nutrition and, as expected, pancreatic development and morphology, and metabolic function will differ.

5. The results of the study are well presented and discussed. However, deficiencies in the nutritional management of yak calves in the control and experimental groups may affect the results obtained.

Author Response

Response to Reviewer 1 Comments

Dear reviewer,

Thanks for your comments and suggestions, I have finished the related revision as followed. Specific modifications are shown in the manuscript animals-1875035.

Point 1: There is a lot of redundant information in the introduction, such as the general functions of the pancreas and the effect of taurine supplementation on production performance and pancreatic development in the broiler.

Response 1: Excess content is omitted from the introduction, as modified in lines 58 to 108.

Point 2: The aim of the study is not sufficiently explained in the introduction.

Response 2: The introduction has been revised, and the purpose of the study is reflected at the end. The specific revisions are shown in lines 58 to 108.

Point 3: The feeding of yak calves in the control group and test group after weaning and the feeding during the experiment were not written clearly enough in the materials and methods section. It's confusing!!

Response 3: The feeding situation in the feeding and management of yak calves has been modified. The specific modifications are shown in lines 110 to 146.

Point 4: Control and test groups should be fed at the same level regarding nutrient content so that the expected effects can be evaluated. I understand that the test group was given concentrated feed in addition to the control group, which indicates that the test group is getting more nutrition and, as expected, pancreatic development and morphology, and metabolic function will differ.

Response 4: At present, yak rearing in the Qinghai plateau of China is still dominated by grazing to a large extent. Before the nutritional requirements of yaks have been met, early supplementation of concentrate after feeding is a great progress. The purpose of this study was to investigate the effect of concentrate supplementation on the development of pancreas in yak calves. Under the condition that the solid dry matter intake of the control group and the experimental group was the same, the purpose of concentrate supplementation was to improve the nutritional level and highlight the more nutrition obtained by the experimental group, so that the morphological development and metabolic function of the pancreas were better. The team also published research in other directions based on supplementary feeding of yak calves. Specific references are as follows.

[1] Guo, W.J., Liu, S.J., Feng, Y.Z., Sun, L., Cui, Z.H. Effects of supplementary feeding during lactation on growth performance, diarrhea frequency and incidence frequency of yak calves. Chinese Journal of Animal Nutrition 2022, 34 (01), 422-431.

[2] Wang, Y., Guo, W.J, Hao, W.J., Cui, Z.H., Liu, S.J. Effects of supplementary feeding during lactation on growth performance, rumen development and microflora of yak calves. Chinese Journal of Animal Nutrition 2022, 34 (05), 3066-3076.

[3] Wang, Y., Xia, H.Z., Yang, Q.E., Yang, D.Y., Liu, S.J., Cui, Z.H. Evaluating Starter Feeding on Ruminal Function in Yak Calves: Combined 16S rRNA Sequencing and Metabolomics. Frontiers in Microbiology 2022, 13.

[4] Cui, Z.H. Effect of yak calving method on growth and digestive tract development. Northwest Agriculture and Forestry University 2020.

[5] Cui, Z.H., Wu, S.R., Li, J.L., Yang, Q.E., Chai, S.T., Wang, L., Wang, X., Zhang, X.W., Liu, S.J., Yao, J.H. Effect of Alfalfa Hay and Starter Feeding Intervention on Gastrointestinal Microbial Community, Growth and Immune Performance of Yak Calves. Frontiers in microbiology 2020, 11, 994.

[6] Wu, S.R., Cui, Z.H., Chen, X.D., Wang, P.Y., Yao, J.H. Changed Caecal Microbiota and Fermentation Contribute to the Beneficial Effects of Early Weaning with Alfalfa Hay, Starter Feed, and Milk Replacer on the Growth and Organ Development of Yak Calves. Animals 2019, 9(11), 921.

Point 5: The results of the study are well presented and discussed. However, deficiencies in the nutritional management of yak calves in the control and experimental groups may affect the results obtained.

Response 5: Feeding management has been modified in the experiment, the specific content is shown in lines 110 to 146.

Reviewer 2 Report

Very good manuscript,but need magor revision. 

1. Table 1: what is means of CP, EE, NDF....

2. Line252:Del:The set thresholds were VIP > 1.0, FC > 1.2, or FC < 0.833, and p-value < 0.05.

3. The abstract needs to be rewritten, and the results need to have specific experimental data.

4. Abbreviations must have full names before they appear of manuscript.

5. The conclusion at the end of the manuscript should also be rewritten. Description of conclusion has no actual content, too vague.

6. References 5,10,11,16,20,51,52,53.... Lack of page Numbers

7.     Moderate English changes required

Author Response

Response to Reviewer 2 Comments

Dear reviewer,

Thanks for your comments and suggestions, I have finished the related revision as followed. Specific modifications are shown in the manuscript animals-1875035.

Point 1: Table 1: what is means of CP, EE, NDF....

Response 1: The abbreviations in the table have been supplemented, and the specific modifications are shown in Table 1 and Table 2.

Point 2: Line252:Del:The set thresholds were VIP > 1.0, FC > 1.2, or FC < 0.833, and p-value < 0.05.

Response 2: The line “the thresholds were VIP>1.0,FC>1.2,or FC<0.833,and p-value<0.05” was deleted, and transferred to “2.4.5. Metabolomics in the pancreas” , and the specific modifications are shown in lines 207 to 208.

Point 3: The abstract needs to be rewritten, and the results need to have specific experimental data.

Response 3: The summary has been rewritten and the changes are detailed in lines 25 to 54.

Point 4: Abbreviations must have full names before they appear of manuscript.

Response 4: All abbreviations in the text and table have been supplemented with full names, and the revisions are shown in the full paper.

Point 5: The conclusion at the end of the manuscript should also be rewritten. Description of conclusion has no actual content, too vague.

Response 5: The conclusion at the end of the manuscript has been rewritten, the details were shown in lines 541 to 550.

Point 6: References 5,10,11,16,20,51,52,53.... Lack of page Numbers

Response 6: I have checked and supplemented the page numbers of the related references, and the specific modifications were shown in lines 569 to 693.

Point 7: Moderate English changes required

Response 7: Moderate English revision have finished in the full article.

Round 2

Reviewer 1 Report

The manuscript is unsuitable for publication in Animals Journal

Reviewer 2 Report

satisfaction for my comments answer! Thank you very much!